# Neuroprotective Effect of Resveratrol against Manganese-Induced Oxidative Stress and Matrix Metalloproteinase-9 in an “In Vivo” Model of Neurotoxicity

**DOI:** 10.3390/ijms25042142

**Published:** 2024-02-10

**Authors:** Tiziana Latronico, Rocco Rossano, Daniela Valeria Miniero, Elisabetta Casalino, Grazia Maria Liuzzi

**Affiliations:** 1Department of Biosciences, Biotechnologies and Environment, University of Bari “A. Moro”, 70126 Bari, Italy; danielavaleria.miniero@uniba.it (D.V.M.); graziamaria.liuzzi@uniba.it (G.M.L.); 2Department of Sciences, University of Basilicata, 85100 Potenza, Italy; rocco.rossano@unibas.it; 3Department of Veterinary Medicine, University of Bari “A. Moro”, 70010 Bari, Italy; elisabetta.casalino@uniba.it

**Keywords:** manganese, central nervous system, neurotoxicity, matrix metalloproteinases, oxidative stress, resveratrol

## Abstract

Chronic exposure to manganese (Mn) leads to its accumulation in the central nervous system (CNS) and neurotoxicity with not well-known mechanisms. We investigated the involvement of matrix metalloproteinase (MMP)-2 and -9 in Mn neurotoxicity in an in vivo model of rats treated through an intraperitoneal injection, for 4 weeks, with 50 mg/kg of MnCl_2_ in the presence or in the absence of 30 mg/kg of resveratrol (RSV). A loss of weight was observed in Mn-treated rats compared with untreated and RSV-treated rats. A progressive recovery of body weight was detected in rats co-treated with Mn and RSV. The analysis of brain homogenates indicated that RSV counteracted the Mn-induced increase in MMP-9 levels and reactive oxygen species production as well as the Mn-induced decrease in superoxide dismutase activity and glutathione content. In conclusion, Mn exposure, resulting in MMP-9 induction with mechanisms related to oxidative stress, represents a risk factor for the development of CNS diseases.

## 1. Introduction

Experimental evidence has suggested that heavy metals play a key role in the induction or exacerbation of several diseases of the central nervous system (CNS) [1]. Manganese (Mn) is an essential element for many physiological cellular processes; however, its overexposure may result in the accumulation in brain tissue, affecting both motor and cognitive functions, thus leading to a neurodegenerative disorder similar to Parkinson’s disease [2,3]. Experimental evidence has indicated that the exposure to Mn has detrimental effects on the brain at any stage of life, determining neuromotor and neurobehavioral deficits as well as neuropsychiatric disorders [4].

Epidemiological and experimental studies have demonstrated that children and adolescents are more vulnerable to Mn-induced neurotoxicity due to brain developing processes and immaturity of excretion mechanisms [5,6]. On the other hand, aging is also a risk factor for Mn-induced neurotoxicity since Mn exposure can result in several cellular and metabolic alterations of an aging brain, which can contribute to the development of age-related neurological diseases [7,8].

It is well documented that manganese can cross the blood–brain barrier (BBB) through both facilitated diffusion and active transport. Furthermore, other transport systems such as metal transport 1 (DMT-1)-mediated transport and ZIP8- and transferrin receptor (TfR)-dependent transport allow its entry into the brain [9]. Among these importers, DMT-1 is the primary transport system for Mn^2+^ while TfR-dependent transport is the primary transporter for Mn^3+^. Experimental studies have demonstrated the increase in DMT-1 protein levels in different rat brain regions with a concurrent increase in Mn levels [10]. The mechanisms of Mn efflux are not fully known. Scientific evidence has highlighted that the dysregulation of influx and efflux mechanisms results in Mn accumulation in the globus pallidus, subthalamic nucleus, substantia nigra, and striatum, causing motor and sensory disorders, as well as neuropsychiatric and cognitive deficits [11].

Astrocytes act as the primary homeostatic regulator of Mn storage in the brain [12]. Nevertheless, high Mn accumulation in astrocytes may decrease glutamate uptake and cause excitotoxicity [13]. At the cellular level, Mn is sequestered into mitochondria where it impairs oxidative phosphorylation and increases matrix calcium levels, resulting in the generation of intracellular reactive oxygen species (ROS) [14,15].

Besides these direct mechanisms, Mn can also indirectly induce neurotoxicity through the activation of glial cells and the release of inflammatory mediators such as matrix metalloproteinases (MMPs) [16,17,18,19]. MMPs constitute a large family of calcium-dependent zinc-containing endopeptidases, which elicit different physiological and pathological functions [20]. In the brain, MMPs are essential for tissue formation, neuronal network remodeling, angiogenesis, and wound healing. Conversely, in pathological conditions, MMPs are involved in various processes including neuroinflammation and neurodegeneration [21]. A high expression of MMPs is well documented in various neurological disorders including Parkinson’s disease (PD), Alzheimer’s disease (AD), and multiple sclerosis (MS) [22,23].

Our research group has previously demonstrated, in an in vitro study, that Mn is able to induce the production of MMP-2 and MMP-9 in astrocytes via mechanisms in part related to the dysregulation of redox homeostasis. Indeed, increased levels of intracellular ROS and a decrease in superoxide dismutase (SOD) activity were found in Mn-treated astrocytes. Furthermore, it has been shown that the antioxidant compound resveratrol (RSV) was able to counteract the increase in MMP-9 expression in Mn-treated astrocytes with mechanisms associated with the reduction in oxidative stress and modulation of the ERK signaling pathway [19].

Based on these results, in this work, we evaluated, in an in vivo rat model, the effects of Mn on MMP-2 and MMP-9 activity and expression, demonstrating that Mn upregulates MMP-9 in plasma and brain homogenates. The increase in ROS levels and the reduced activity of the endogenous antioxidants SOD and GSH suggest that the overexpression of MMP-9 is related to a dysregulation of the intracellular redox homeostasis. To assess a possible therapeutic efficacy of RSV, we also measured the same parameters in rats co-treated with both Mn and RSV and demonstrated the ability of RSV to counteract Mn-induced neurotoxicity with mechanisms related to the reduction in oxidative stress.

## 2. Results

### 2.1. Mn Induces Locomotor Impairment in Rats 

Figure 1 reports the mean score obtained through the evaluation of the locomotor status of rats measured during the treatment period. As shown, in rats treated with Mn, clinical signs began to appear in the second week with a gradual worsening in the third and fourth weeks of treatment. 

Co-treatment with Mn and RSV resulted in an improved clinical status; indeed, clinical signs began to manifest from the third week of treatment with a lower median score in comparison with rats treated with Mn, without any worsening in the fourth week of treatment. No evident signs of locomotor disability were evidenced in the groups of both untreated rats and RSV-treated rats. Table 1 reports the number of rats assigned to each score during the period of treatment. 

### 2.2. Variation of Rat Body Weight in the Different Groups of Rats during the Treatment Period

To evaluate whether Mn treatment impaired development of animals, the rat body weight was measured once a week during the treatment period. As shown in Figure 2a, during the 4 weeks of treatment, a progressive increase in body weight was observed in the control groups represented by untreated (CTRL) and RSV-treated rats (RSV). In contrast, treatment with Mn determined a significant and progressive loss of weight during the treatment period in comparison with the weight measured at the beginning of the treatment. A lower loss of body weight was evident in rats co-treated with Mn and RSV (Mn + RSV) in comparison with Mn-treated rats. As shown in Figure 2b, at the end of the treatment, a statistically significant loss of body weight was observed in the groups of Mn-treated and Mn + RSV co-treated rats in comparison with control rats. In contrast, no difference in body weight was observed between RSV-treated and CTRL rats.

### 2.3. Mn Increases Levels of MMP-9 in Plasma

MMP-2 and MMP-9 levels were detected by gelatin zymography in the plasma of the different groups of rats collected at the end of treatment. As shown in the representative zymographic gel (Figure 3a), four bands of digestion, corresponding to activated MMP-2 (67 kDa), pro-MMP-2 (72 kDa), activated MMP-9 (87 kDa), and pro-MMP-9 (92 kDa), were observed in the plasma samples from the different categories of rats. The quantitative analysis of total MMP-2 and MMP-9 levels, calculated after scanning densitometry and computerized analysis as a sum of the pro- and activated forms (Figure 3b), indicated that MMP-9 plasma levels were increased in Mn-treated rats in comparison with the other analyzed groups. In contrast, levels of MMP-2 did not show any significant variation in all groups of rats analyzed.

### 2.4. Mn Increases Levels and mRNA Expression of MMP-9 in Rat Brain

Differently from the results obtained in plasma samples, the zymographic analysis of the brain extracts enriched in gelatinases showed only one band of digestion, corresponding to MMP-9 and trace amount of MMP-2 (Figure 4a). In particular, low levels of MMP-9 were present in the untreated (CTRL) and RSV-treated groups. A statistically significant increase in MMP-9 levels was observed in Mn-treated rats in comparison with CTRL rats. As observed in rats co-treated with Mn and RSV, the Mn-induced increase in MMP-9 was counteracted by RSV (Figure 4b).

The investigation of MMP expression indicated that MMP-2 and MMP-9 mRNA levels were statistically increased in brain extracts from Mn-treated rats in comparison with CTRL rats and this increase was counteracted by treatment with RSV (Figure 4 c,d).

### 2.5. Mn Induces the Production of ROS in Brain Homogenates

To study whether the increase in MMP-9 levels and expression induced by Mn could be related with the oxidative stress, ROS levels were assayed in brain homogenates from the different groups of rats. As shown in Figure 5, a statistically significant increase in ROS production was observed in brain homogenates of Mn-treated rats in comparison with CTRL. In contrast, a statistically significant reduction in ROS production was observed in the group of rats co-treated with Mn and RSV in comparison with Mn-treated rats.

### 2.6. Mn Affects the Brain Redox State by Reducing Glutathione (GSH) Levels and Superoxide Dismutase (SOD) Activity

To better elucidate the mechanisms by which Mn alters the redox state in the brain, we evaluated its effect on two endogenous antioxidant defense systems: the non-enzymatic GSH and the enzymatic SOD.

As shown in Figure 6a, a statistically significant decrease in GSH levels (a) and SOD activity (b) was observed in brain homogenates from the group of Mn-treated rats. In contrast, the co-treatment with RSV significantly increased GSH levels and SOD activity to levels comparable to those observed in the CTRL group.

## 3. Discussion

Manganese (Mn) is an essential element for living organisms, but excessive exposure to this metal represents a major public health concern. Particularly in the occupational setting, Mn-induced toxicity occurs in miners, ferroalloy workers, battery manufacturers, and car mechanics due to its widespread use in manufacturing of dry batteries, gasoline additives, and fungicides [24]. Furthermore, Mn exposure may also result from a total parenteral nutrition, where an excessive dose may prematurely induce neurological disorders in children [25,26,27].

Experimental evidence has revealed that chronic or acute exposure to high concentrations of Mn reliably leads to pathological processes characterized by irreversible central nervous system (CNS) damage [28,29]. Although the mechanisms of Mn neurotoxicity are not fully known, it has been clearly defined that the brain structures primarily affected by Mn intoxication include the striatum, globus pallidus, and substantia nigra [30,31,32]. Indeed, the overdose accumulation of Mn in these specific brain areas triggers neurotoxicity, causing neurological syndromes similar to chronic Parkinson’s disease (PD) and manganism [1,33].

In this study, by using an animal model represented by rats treated with Mn, we demonstrated that Mn exposure induced a progressive loss of body weight. This loss of weight was not associated with a reduced intake of food since no differences were observed in food consumption between the different groups of rats during the experimental period. As already suggested by other authors, the progressive weight loss in Mn-treated rats could be associated with Mn-induced metabolic and biochemical dysfunctions such as deficiency in energy metabolism and/or alterations in the functionality of the hypothalamus, which is responsible for the control of body weight [25,34,35,36,37]. In another study, Cannon et al. [38], using an animal model of PD, showed that weight loss was associated with decreased motor activity and gait disturbances. In our study, Mn treatment markedly induced locomotor disturbances beginning on study day 15 with a progressive worsening until the sacrifice. As assessed, the observed symptoms were not due to any problem with the method of Mn administration, but rather due to the effect of Mn intoxication, since no motor disturbances were observed in the groups of RSV-treated or untreated rats. Mn-induced locomotor deficits in humans [39,40] and laboratory animals were documented in numerous studies [41,42]. Hypotheses to explain motor deficits point to the dysregulation of monoaminergic neurotransmission with consequent impairment of the functionality of the globus pallidus and striatum. 

In the brain, an excess of Mn accumulates preferentially in astrocytes, affecting their ability to indirectly induce impairment of normal neuronal function and inducing an exaggerated production of inflammatory and neurotoxic factors [43].

The analysis of plasma collected at the end of the treatment indicated that Mn induced increased levels of MMP-9, whereas it had no effect on MMP-2, whose levels are generally constant in plasma. Similarly, an increase in MMP-9 levels was also evident in brain homogenates from Mn-treated rats. In the homogenate samples, MMP-2 was present only in trace amounts, making it difficult to quantify its levels. However, we cannot exclude that MMP-2 levels are also increased in brain homogenates from Mn-treated rats, since the mRNA expression analysis of brain homogenates showed a significant increase in both MMP-2 and MMP-9. 

MMP-2 and MMP-9 are involved in numerous pathophysiological processes including CNS injuries and diseases [44,45,46]; therefore, the increase in levels and mRNA expression of both MMPs in the brain of rats subjected to Mn exposure could be an important mechanism by which this metal exerts neurotoxicity. Mechanisms by which Mn induce MMP expression are not fully elucidated, but it has been shown that in mixed glial cells, the Mn-induced production of inflammatory mediators was associated with the activation of the MAPK transduction pathway, resulting in the activation of NF-kB and activator protein-1 (AP-1) [47]. In vivo studies have confirmed that some of the mechanisms of inflammation are modulated by Mn and, particularly, the Mn-induced toxicity in mice was associated with an increased activation of NF-kB [48]. It is noteworthy that NF-kB plays a central role in the inflammatory response by regulating the gene expression of different neurotoxic mediators such as pro-inflammatory cytokines and MMPs [49]. Therefore, the increase in MMP-2 and MMP-9 observed in this study could be a consequence of NF-kB activation. 

Several experimental pieces of evidence have suggested that ROS are key regulators of MMP production and that oxidative stress is a primary mechanism of Mn-mediated neurodegeneration [50,51,52,53]. In this study, we investigated whether in Mn-treated rats the increase in MMP levels and expression could be related to an alteration of the redox state in the brain. As a result, increased ROS levels and decreased SOD activity were found in brain homogenates of Mn-treated rats. It is well known that SOD is a major antioxidant enzyme that, due to its ability to remove ROS, protects cells from the oxidative damage of many biological macromolecules, such as proteins, lipids, and DNA. Therefore, the decreased activity of this enzyme observed in this study may be responsible for the increase in free radicals, which in turn stimulate the production of toxic factors. This condition of oxidative stress is further confirmed through the observation that GSH levels are reduced in brain homogenates from Mn-treated rats. GSH is one of the most important endogenous antioxidants and it is essential for maintaining intracellular redox homeostasis. A reduction in its levels may, therefore, contribute to the increase in ROS production, since in the state of severe oxidative stress, GSH is converted into the oxidized form (GSSG), leading to its accumulation within the cytosol. In such a situation, the use of exogenous antioxidants may help to counteract an inefficient endogenous defense system, thus reducing the cumulative effects of ROS-mediated oxidative damage [54]. Between the natural antioxidants, the dietary polyphenols have attracted great interest in the treatment of neurological diseases given their documented anti-inflammatory, immunomodulatory, antioxidant, and neuroprotective properties [55,56,57]. In this study, we focused our attention on resveratrol (RSV). RSV is a polyphenolic compound found in a large number of plant species including mulberries, peanuts, grapes, and red wine. Various scientific reports have shown that RSV offers protective effects against a number of cardiovascular and neurodegenerative diseases and cancer [58,59,60]. Other than being a potent antioxidant, RSV possesses multitargeting biological effects, such as the reduction in oxidative stress, the promotion of cell growth, the inhibition of brain pro-inflammatory responses, the prevention of neuronal cell death, and more [61]. Our research group previously demonstrated that RSV modulates the expression and activity of MMP-2 and MMP-9 in both LPS and Mn-treated astrocytes [19,62]. Similarly, in this study, we demonstrated that RSV is able to counteract the locomotor disability observed in Mn-treated rats, and the induction of both MMP-2 and MMP-9 expression in the brain from rats co-treated with Mn. The mechanism by which RSV exerts this effect is not yet clear, but we can suggest that RSV downregulates the expression of MMP-2 and MMP-9 at the transcription level, downregulating various signaling pathways that mediate the induction of MMPs [63]. Experimental evidence has suggested that RSV modulates the activity of enzymes and transcription factors that influence gene expression. At the molecular level, RSV downregulates nuclear factors, such as NF-kB, which plays a central role in cellular signaling cascades, regulating DNA transcription and modulating the gene expression of different inflammatory molecules such as MMPs [64]. In this respect, in an in vitro study, we previously demonstrated that the mechanism by which RSV significantly counteracts the increase in MMP-9 expression in Mn-treated astrocytes is associated with the downregulation of ERK1/2, which represents the main signaling pathway responsible for MMP-9 transcription [19].

As a result of this study, we also demonstrated the ability of RSV to counteract the generation of ROS and to increase the activity of SOD and GSH content in Mn co-treated rats. Considering that Mn is also indirectly involved in the induction of MMPs through the generation of ROS, we cannot exclude that the inhibition of MMP-2 and MMP-9 expression by RSV is a direct consequence of the reduction in ROS production. This finding supports the hypothesis that the effect of Mn on the upregulation of MMP-2 and MMP-9 gene expression could be attributed to an altered balance between the endogenous antioxidant systems and ROS production and that RSV is able to restore this imbalance, counteracting Mn action. On the other hand, we cannot exclude that the neuroprotective action of RSV may be related not only to its ability to counteract Mn-induced ROS generation, but also to its inhibitory effect on MMPs through the modulation of transcription factors.

## 4. Materials and Methods

### 4.1. Chemicals

Gelatin, RSV and MnCl_2_, 5,5′-dithiobis(2-nitrobenzoic acid (DTNB), and the GSH standard were provided by Sigma (St. Louis, MO, USA). 2′,7′-Dichlorofluorescein was from Calbiochem (La Jolla, CA, USA). The Bradford reagent, standard proteins, and R-250 Coomassie Brilliant Blue were purchased from Bio-Rad (Hercules, CA, USA). Gelatin Sepharose 4B was from Pharmacia Biotech (Uppsala, Sweden). Primer pairs specific for MMP-2, MMP-9, and glyceraldehyde 3-phosphate dehydrogenase (GAPDH) were from Sigma Genosys (Cambridgeshire, UK). An RNeasy mini kit was from Qiagen (Valencia, CA, USA). All the reagents for RT-PCR were purchased from Invitrogen (San Diego, CA, USA).

### 4.2. Ethics Statement

All experimental procedures involving animals were carried out in strict accordance with the recommendations in the NIH Guide for the Care and Use of Laboratory Animals and approved by the Institutional Animal Care and Use Committee of University of Bari Aldo Moro, Italy (Permit Number: 23-98-A). All efforts were made to minimize the number of animals used and to ameliorate their suffering.

### 4.3. Experimental Design

A total of 28 male Wistar rats (Harlan, Italy) of 16 weeks of age (body weight: 375 ± 70 g) were randomly divided into four groups of 7 rats each and treated twice a week, for 4 weeks, through an intraperitoneal injection with (1) 30% ethanol (CTRL); (2) RSV (30 mg/kg) in 30% ethanol; (3) MnCl_2_ (50 mg/kg) in 30% ethanol; (4) MnCl_2_ (50 mg/kg) plus RSV (30 mg/kg) in 30% ethanol. The dose of Mn was chosen based on several literature data showing a significant increase in Mn accumulation in brain tissues and alteration of the biochemical properties [37,65]. The weight of the animals was monitored once a week throughout the experimental period and the dose of Mn was adjusted before every injection, taking into account the weight variation. 

During the experimental period, rats were housed in clear plastic cages, in the animal facility of the Department of Biosciences, Biotechnologies and Environment of University of Bari (Italy). All animals were maintained at the room temperature of 22 °C, with humidity at 40–50%, on a 12 h:12 h light/dark cycle with free access to tap water and fed with a commercial standard pellet diet (2018 Teklad Global 18% Protein Rodent Diet) (Envigo Harlan, Indianapolis, IN, USA).

### 4.4. Clinical Observation

During the treatment period, animals were observed once a week for locomotor status assessment according to the following scale: 0 = no sign (there is no observable difference from non-treated animals); 1 = loss of tail tonicity or hind limb weakness; 2 = moderate paraparesis (one hind limb is paralyzed); 3 = severe paraparesis (complete hind limb paralysis with inability to maintain posture or walk); 4 = moribund status. Accurate grading was verified by scoring the animalsby three different investigators blind to the treatment given to the animals. At the end of each week, for each group of rats, the mean score was calculated by applying the following formula: $$\mathrm{mean}\;\mathrm{score}=\frac{\sum fx}{\sum f}$$
where *f* frequency that represents the number of times a score occurs, and *x* is the matching score.

### 4.5. Collection of Brain and Blood Samples

As reported in [66], at the end of the treatment period, blood samples were collected from the rat tails in tubes containing heparin (50 U/mL of blood) after cleaning of the skin. Plasma, obtained after centrifugation at 1800 rpm for 10 min at RT, was recovered, aliquoted, and stored at −80 °C until use. Then, the treated and control rats were euthanized via exposure to carbon dioxide and sacrificed by decapitation. The brains were immediately removed from the skull, weighed, and stored at −80 °C.

### 4.6. Preparation of Brain Homogenates

Frozen brains from untreated rats or from rats treated with Mn or RSV or co-treated with Mn and RSV were homogenized at 4 °C with a working buffer (50 mM Tris-HCl, 5 mM CaCl_2_, 150 mM NaCl, 0.05% Brij-35, 1% Triton-X-100, 100 mM PMSF, pH 7.5) using a Potter-Elvehjem homogenizer. Aliquots of brain homogenates were then stored at −80 °C until use. An aliquot of each brain homogenate was incubated on ice for 60 min and centrifuged at 12,000× *g* for 30 min at 4 °C and supernatants obtained were used for MMP-2 and MMP-9 purification by affinity chromatography. Protein concentration in brain homogenates and supernatants was determined using the Bradford assay [67].

### 4.7. Purification of MMP-2 and MMP-9 by Affinity Chromatography

To improve the ratio of MMP-2 and MMP-9 versus total proteins, supernatants obtained from brain homogenates were subjected to a miniaturized affinity chromatography purification step on Gelatin Sepharose 4B, according to the method described by Zhang and Gottschall [68].

Briefly, supernatants were loaded onto the Gelatin Sepharose 4B, packed in columns (3.3 µL of resin/mg of supernatant proteins), and incubated at 4 °C for 30 min. After pass-through removal, the column was washed with a working buffer (50 mM Tris-HCl, 5 mM CaCl_2_, 150 mM NaCl, 0.05% Brij-35, 100 mM PMSF, pH 7.5) and MMPs were then eluted with a working buffer containing 10% dimethyl sulfoxide (DMSO). The brain extracts obtained, enriched in gelatinases, were aliquoted and stored at −80 °C until use.

### 4.8. Detection of MMP-2 and MMP-9 by Zymography

MMP-2 and MMP-9 activity in brain extracts and plasma was detected by SDS-PAGE zymography according to Di Bari et al. [69]. 

Briefly, 50 μL of brain extracts enriched in gelatinases, corresponding to about 30 μg of proteins, or 0.4 µL of plasma, was supplemented with 10 μL of an electrophoresis non-reducing loading buffer: 4% (*w*/*v*) SDS, 12% (*w*/*v*) glycerol, 0.01% (*w*/*v*) bromophenol blue, 50 mM Tris–HCl (pH 6.8). Samples were then analyzed in 7.5% polyacrylamide gels copolymerized with 0.1% (*w*/*v*) gelatin. After electrophoresis, gels were washed in a buffer containing 2.5% (*w/v*) Triton X-100, then incubated for 24 h at 37 °C in a developing buffer. MMP-2 and MMP-9 activity was detected as a white band on a blue background of gels stained with Coomassie Brilliant Blue R-250. MMP activity was expressed as optical density (OD) × mm^2^, after scanning densitometry and quantification by computerized image analysis using the Image Master 1D program (Pharmacia Biotech, Uppsala, Sweden).

### 4.9. Reverse Transcription–Polymerase Chain Reaction

The mRNA expression of MMP-2 and MMP-9 was detected as reported in [19]. Briefly, 500 ng of RNA extracted from brain homogenates was used to synthetize complementary DNA (cDNA). A total of 25 ng of reverse transcription products was amplified in each reaction with 25 cycles of the polymerase chain reaction (PCR) using rat primers specifically designed for MMP-2 and MMP-9 genes. 

Additionally, 1.5% agarose gels were used to analyze the PCR products stained with ethidium bromide; then, gels were subjected to a densitometric analysis as described for protein gels. Levels of the mRNA expression of MMP-2 and MMP-9 were normalized to GAPDH expression, a housekeeping gene used as an internal control.

### 4.10. Detection of Reactive Oxygen Species

The detection of reactive oxygen species was performed as reported by Latronico et al. [70]. Briefly, aliquots of brain homogenates, corresponding to 30 µg of total proteins, were loaded with 50 μM 2′,7′-dichlorofluorescein in PBS and incubated at 37 °C for 30 min. The fluorescent signal of oxidized 2′,7′-dichlorofluorescein was measured (excitation: 485 mm; emission: 520 nm) using a fluorometer (Perkin Elmer LS50, v.6). Results were expressed as a percentage to CTRL, arbitrarily set at 100%.

### 4.11. Detection of SOD Activity 

SOD activity in 10 µg of brain homogenates was determined spectrophotometrically from its ability to inhibit the autoxidation of epinephrine as reported by Latronico et al. [19].

### 4.12. Estimation of Reduced Glutathione Levels (GSH)

GSH content was measured according to the method of Moron et al. [71]. Briefly, 0.1 mL of brain homogenates was resuspended in 10% trichloroacetic acid and centrifuged to precipitate the proteins. Then, a 0.2 M phosphate buffer and 0.6 mM DTNB [(5,5′-dithiobis (2-nitrobenzoic acid)] in 0.2 M sodium phosphate, pH 8.0, were added to this homogenate. Briefly, each reaction consisted of 0.6 mM DTNB in 0.2 M sodium phosphate, pH 8.0, brain homogenates (0.1 mL), and a 0.2 M phosphate buffer. The absorbance was measured at 412 nm, and activity was calculated based on a calibration curve plotted using the GSH standard. Results are reported as GSH content/100 µg of the brain homogenate expressed as a percentage of CTRL, arbitrarily set at 100%.

### 4.13. Statistical Analysis 

The statistical analysis was performed on data expressed as mean values ± standard deviation (SD). Values were compared using a One-Way Analysis of Variance (ANOVA) followed by Tukey’s Multiple Comparison Test to assess the statistical significance between the different analyzed groups. The asterisks represent statistically significant values (* = *p* < 0.05; ** = *p* < 0.01). Data were analyzed with GraphPad Prism 5.0 (GraphPad Software, San Diego, CA, USA).

## 5. Conclusions

Results of this study indicate that the dysregulation of factors such as MMP-9, SOD, and GSH might contribute to potentiate the Mn-induced neuroinflammatory pathways, providing new insights into the mechanisms of metal-induced neurodegeneration. Furthermore, the inhibition of MMP-9 expression and the recovery of SOD activity and GSH content in rats co-treated with Mn and RSV demonstrate the potential of exogenous antioxidants for the prevention and the complementary treatment of diseases in which oxidative stress and neuroinflammation play a critical role.

## Figures and Tables

**Figure 1 ijms-25-02142-f001:**
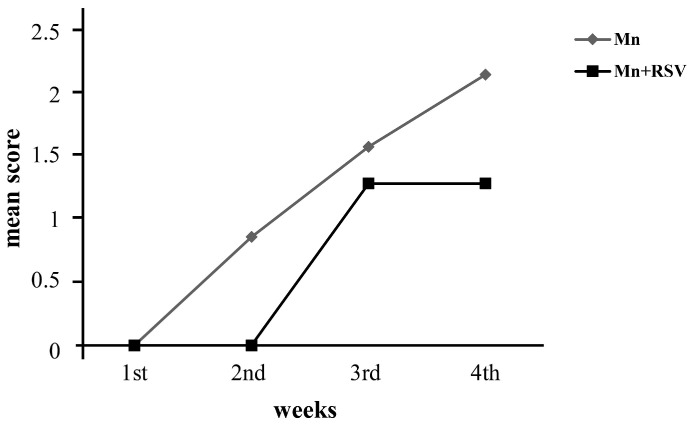
Mean clinical score in the groups of rats treated with Mn or co-treated with Mn and RSV. Rats treated with Mn (*n* = 7) or co-treated with Mn and RSV (Mn + RSV) (*n* = 7) were screened weekly for the presence of signs of motor disability, as reported in the Materials and Methods section. As shown by the mean clinical score, the first signs of motor disability developed earlier in the group of rats treated with Mn than in Mn + RSV co-treated rats and rapidly worsened thereafter. In contrast, a delay in the onset and progression of motor disability was observed in Mn + RSV co-treated rats.

**Figure 2 ijms-25-02142-f002:**
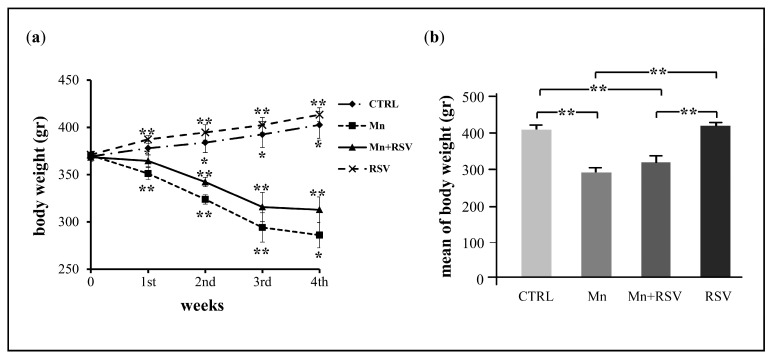
Evaluation of body weight in the different groups of rats during (**a**) and at the end (**b**) of the treatment period. The graph in (**a**) represents the mean ± SD of body weight of rats in the analyzed groups (*n* = 7 per group), monitored once a week. Asterisks represent statistically significant values compared to mean body weight of rats at the beginning of treatment. The histograms in (**b**) represent the means ± SD of body weight of untreated rats (CTRL), Mn-treated rats (Mn), rats co-treated with Mn and resveratrol (Mn + RSV), and RSV-treated rats (RSV), assessed at the end of treatment. The asterisks represent statistically significant values between the different analyzed groups (One-Way ANOVA followed by Tukey’s Multiple Comparison Test; * *p* ≤ 0.05 and ** *p* ≤ 0.001).

**Figure 3 ijms-25-02142-f003:**
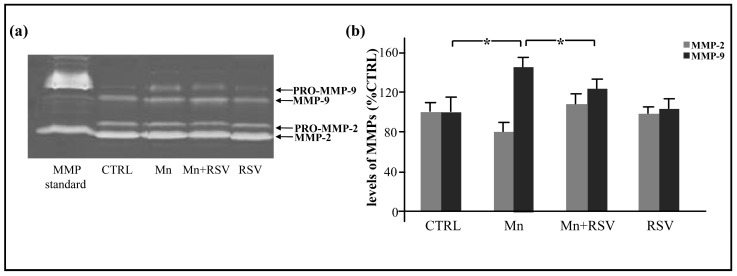
MMP-2 and MMP-9 levels in plasma from the different groups of rats. Representative gel in (**a**) shows MMP-2 and MMP-9 levels, as identified by their apparent molecular mass of 67 and 87 kDa, respectively, using prestained molecular weight markers (Bio-Rad, Hercules, CA, USA). Bands of apparent molecular mass of 72 kDa and 92k Da were also observed, corresponding, respectively, to the pro- forms of MMP-2 and MMP-9. Histograms in (**b**) represent the means ± SD of total MMP-2 and MMP-9 plasma levels from the different groups of rats (*n* = 7 rats/group), expressed as percent of CTRL, after scanning densitometry and computerized analysis of gels. Statistically significant increase between the different analyzed groups is indicated by asterisks (One-Way ANOVA followed by Tukey’s Multiple Comparison Test; * *p* ≤ 0.05).

**Figure 4 ijms-25-02142-f004:**
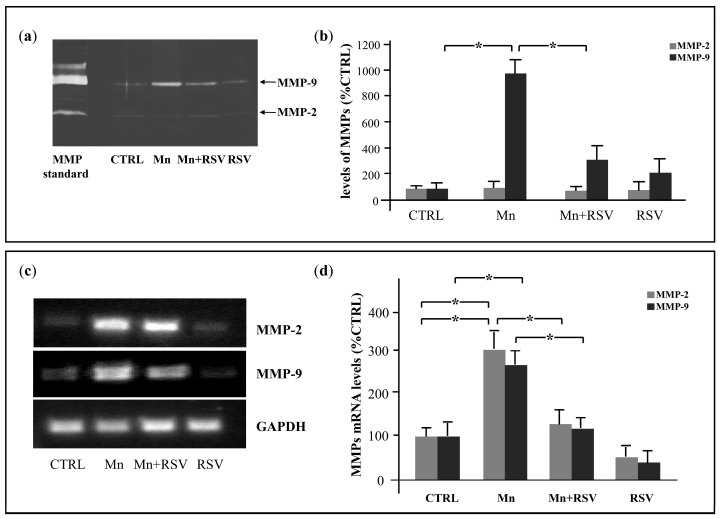
Effect of Mn and RSV on MMP-2 and MMP-9 levels and mRNA expression in brain homogenates. Representative zymographic gel in (**a**) shows MMP-2 and MMP-9 detected in supernatants from brain homogenates from the different groups of rats (*n* = 7/group) after enrichment with miniaturized affinity chromatography purification step on Gelatin Sepharose. Histograms in (**b**) represent the means ± SD of MMP-2 and MMP-9 levels expressed as percent of CTRL, after scanning densitometry and computerized analysis of gels. (**c**,**d**) Representative agarose gels in (**c**) show MMP-2, MMP-9, and GAPDH expression in brain homogenates from the different groups of rats. Histograms in (**d**) represent the means ± SD of MMP-2 and MMP-9 mRNA levels in brain homogenates, expressed as percent of CTRL, after scanning densitometry. Asterisks indicate statistically significant differences in comparison with Mn-treated rats (One-Way ANOVA followed by Tukey’s Multiple Comparison Test; * *p* ≤ 0.05).

**Figure 5 ijms-25-02142-f005:**
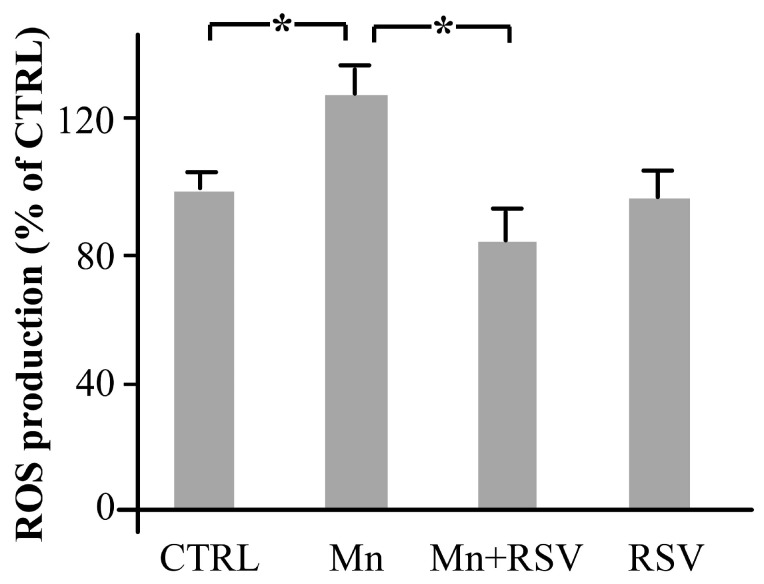
Reactive oxygen species (ROS) production in brain homogenates from the different groups of rats. The presence of ROS was assayed by measuring the changes in the fluorescent signal of 2′,7′-dichlorofluorescein as reported in Materials and Methods section. Histograms represent the means ± SD of ROS production in brain homogenates, expressed as percentage of CTRL, arbitrarily set at 100%. The asterisks represent statistically significant values between the different analyzed groups (One-Way ANOVA followed by Tukey’s Multiple Comparison Test; * *p* ≤ 0.05).

**Figure 6 ijms-25-02142-f006:**
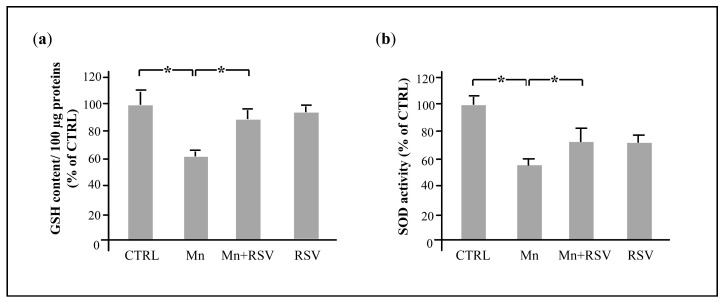
Effect of treatment with Mn and RSV on endogenous antioxidant defense systems. Reduced glutathione (GSH) content and SOD activity in brain homogenates were assayed by spectrophotometric measurement as described in Materials and Methods section. The histograms in (**a**) represent the mean ± SD of GSH levels, related to 100 µg of proteins, expressed as percentage of CTRL. The histograms in (**b**) represent the mean ± SD of SOD activity, measured in 10 μg of brain homogenates and expressed as percentage of CTRL, which was arbitrarily set at 100%. The asterisks represent statistically significant values between the different analyzed groups (One-Way ANOVA followed by Tukey’s Multiple Comparison Test; * *p* ≤ 0.05).

**Table 1 ijms-25-02142-t001:** Evaluation of clinical locomotor status of rats treated with Manganese (Mn) or co-treated with Manganese and Resveratrol (Mn + RSV).

	Rats Treated with Mn					Rats Treated with Mn + RSV				
	Score 0	Score 1	Score 2	Score 3	Score 4	Score 0	Score 1	Score 2	Score 3	Score 4
**1st week**	7	0	0	0	0	7	0	0	0	0
**2nd week**	4	0	3	0	0	7	0	0	0	0
**3rd week**	3	0	1	3	0	2	1	4	0	0
**4th week**	2	0	1	3	1	2	1	4	0	0

The score represents the locomotor status of rats evaluated according to the following scale: 0 = no sign (there is no observable difference from non-treated animals); 1 = loss of tail tonicity or hind limb weakness; 2 = moderate paraparesis (one hind limb is paralyzed); 3 = severe paraparesis (complete hind limb paralysis with inability to maintain posture or walk); 4 = moribund status.

## Data Availability

All data generated or analyzed during this study are included in this article. The datasets used and/or analyzed during the current study are available from the corresponding author on reasonable request.

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
