# Peer review of "Neuroprotective Effect of Resveratrol against Manganese-Induced Oxidative Stress and Matrix Metalloproteinase-9 in an “In Vivo” Model of Neurotoxicity"

_ijms, 2024, doi:10.3390/ijms25042142_

Round 1

Reviewer 1 Report

Comments and Suggestions for Authors

The manuscript entitled"Neuroprotective Effect of Resveratrol against Manganese-In- 2 duced Oxidative Stress and Matrix Metalloproteinase-9 in a “in 3 vivo” Model of Neurotoxicity" is scientifically good. However, there are some points to be addressed by the authors before it can be accepted.

Comments:

In the materials and methods section, the experimental design author should mention what type of diet was fed to rats and from where the rat diet was purchased.

In the methods section  Collection of brains and blood samples author should mention as per which reference methodology this procedure was followed. Citation needed.

Throughout the manuscript some chemical formula like CACl2, H2O,NaHCO3/Na2CO is not as per scientific format. 

Why the authors have not included any Histopathological studies of before and after treatment and brain histology changes associated with it?

Manuscript needs technical English correction.

Comments on the Quality of English Language

The manuscript needs technical English correction.

Author Response

Reviewer #1

Comment: In the materials and methods section, the experimental design author should mention what type of diet was fed to rats and from where the rat diet was purchased.

Response:

The type of diet used to feed the rats has been mentioned in the materials and methods section, paragraph “Experimental design”, pag 9, lines 347-349.

Comment: In the methods section “Collection of brains and blood samples” author should mention as per which reference methodology this procedure was followed. Citation needed.

Response:

In the methods section “Collection of brains and blood samples”, the paragraph has been modified and reference 65 has been added according to the request of the reviewer (pag. 10, lines 364-369).

Comment: Throughout the manuscript some chemical formula like CACl2, H2O, NaHCO3/Na2CO is not as per scientific format.

Response:

We have corrected the chemical formula of CaCl2 but we have not found the chemical formula of both H2O and NaHCO3/Na2CO throughout the manuscript.

Comment: Why the authors have not included any histopathological studies of before and after treatment and brain histology changes associated with it?

Response:

We could not perform any histopathological studies before and after the treatment and brain histological changes associated with it, because we didn’t have enough material since most of the brain was used for the purification of MMP-2 and MMP-9 by affinity chromatography.

Comment: Manuscript needs technical English correction.

Response:

According to the request of the reviewer the English has been revised.

Reviewer 2 Report

Comments and Suggestions for Authors

Authors described the resveratrol improved the locomotor status of MnCl2-induced rats.  Some papers had reported the similar functions of resveratrol in MnCl2-induced models.  No new information was found in the present MS. It was noted that the body weights of rats in MnCl2-induced models showed significantly body weight loss (less than 75% of the control in the four-week treatments), and MnCl2 induction also showed systemic toxicities in differnt organs and tissues. Therefore, it was not a general models for neuron protections. 

1. Resveratrol attenuates manganese-induced oxidative stress and neuroinflammation through SIRT1 signaling in mice. Food Chem. Toxicol.  2021, 153, 112283

2. Resveratrol Attenuated Manganese-Induced Learning and Memory Impairments in Mice Through PGC-1Alpha-Mediated Autophagy and Microglial M1/M2 Polarization. Neurochem. Res. 2022, 47, 3414-3427.

Author Response

Referee #2
Comment: Authors described the resveratrol improved the locomotor status of MnCl2 induced rats. Some papers had reported the similar functions of resveratrol in MnCl2 induced models. No new information was found in the present MS. It was noted that the body weights of rats in MnCl2 induced models showed significantly body weight loss (less than 75% of the control in the four week treatments), and MnCl2 induction also showed systemic toxicities in differ e nt organs and tissues. Therefore, it was not a general models for neuron protections.

1. Resveratrol attenuates manganese induced oxidative stress and neuroinflammation through SIRT1 signaling in mice. Food Chem. Toxicol. 2021, 153, 112283
2. Resveratrol Attenuated Manganese Induced Learning and Memory Impairments in Mice Through PGC-1Alpha Mediated Autophagy and Microglial M1/M2 Polarization. Neurochem. Res. 2022, 47, 3414 3427.

Response:
1. According to the observation of the referee, we cannot exclude that MnCl2 might induce systemic toxicities in different organs and tissues. However, we do not agree with the referee that the model we used in this study does not represent a good model of neurotoxicity , since it has been used in different studies Ref . 36, 40, 41 of the manuscript; Fan et al 2020 doi: 10.1007/s11064 020 03059 2 )). On the other hand, the same papers cited by the referee in his comment refer to models similar to ours that the authors have used to
evaluate the protective effects of resveratrol (RSV) on Mn induced oxidative stress and neuro inflammation

2. The main aim of our study was to evaluate the neuroprotective effect of RSV on MMP 2 and MMP 9 activity and expression. To the best of our knowledge, there are no other studies carried out on in vivo models of rats treated with Mn that studied how RSV treatment relieves oxidative stress and neuroinflammation using MMPs as targets. We therefore believe that this is an original study.
On the other hand, MMP 2 and MMP 9 have been indicated for several decades as important pathogenetic factors in neuroinflammation and oxidative stress (Ref. 19 22 of the manuscript). In addition, this study represents the natural continuation of our previo us in vitro study (ref. 18 of the manuscript), in which, by using an astrocyte cell model, we demonstrated that Mn neurotoxicity also derives from its ability to induce the expression of MMP 9.

Reviewer 3 Report

Comments and Suggestions for Authors

This manuscript presents the neuroprotective effect of Resveratrol (RSV) against Mn-induced oxidative stress and MMPs. the manuscript needs major revision for the publication in International Journal of Molecular Sciences. Detailed comments are as below.

1.      The authors present % of rats in each week and each score in Table 1. But, since the authors tested only 7 rats, it is not necessary to show the numbers of rats in %. The number of rats is enough. Also, what about the results from control rats which was not treated either Mn or RSV?

2.      The number of Figures should be fixed. For example, Figure 2 is now Figure 1 in the manuscript.

3.      In Figure 1, error bars should be presented.

4.      In Figure 2, the authors showed the changes of body weights of rats. The causes of loss of changes of body weights should be discussed in detail. Moreover, in Figure 2b, it is not clear when the data was obtained. What age of rats?

5.      In Figure 3, Figure 3a and 3b do not match.

6.      Please fix the figure caption for Figure 2 and Figure 3.

7.      In Figure 3, 4, 5, and 6, the error bars for the control samples should be presented.

Author Response

Reviewer 3

Comment: The authors present % of rats in each week and each score in Table 1. But, since the authors tested only 7 rats, it is not necessary to show the numbers of rats in %. The number of rats is enough. Also, what about the results from control rats which was not treated either Mn or RSV?

Response:

  1. We agree with the referee that the number of rats is small but the indication of the percentages in Table 1 emphasizes the impact of treatment on the health status of rat population. Therefore, we would prefer to leave them in table 1.
  2. As already assessed in the Discussion section, pag 7, lines 234-235, no motor disturbances were observed in the groups of RSV-treated or untreated-rats. However, the sentence: “No evident signs of locomotor disability were evidenced in the groups of both untreated rats and RSV-treated rats” was added also in the results section (paragraph 2.1, pag.2, lines 90-91) .

Comment: The number of Figures should be fixed. For example, Figure 2 is now Figure 1 in the manuscript.

Response:

We have now corrected the number of Figure 2 in the caption.

Comment: In Figure 1, error bars should be presented.

Response:

Error bars have been added to Figure 1

Comment: In Figure 2, the authors showed the changes of body weights of rats. The causes of loss of changes of body weights should be discussed in detail. Moreover, in Figure 2b, it is not clear when the data was obtained. What age of rats?

Response:

  1. Hypotheses to explain the causes of body weight loss or changes were discussed in the "Discussion" section, pag. 7, lines 226-231).
  2. Both in the text (Paragraph 2.2, pag. 3, line 113,) and in the caption of Figure 2 it is indicated that the data reported in the histograms of Figure 2b represent the mean of rat body weight measured at the end of the treatment. Considering that at the beginning of the treatment the rats were 16 weeks old and that the treatment lasted for 4 weeks, at the end of the treatment the age of the rats was 20 weeks.

Comment: Results reported in fig 3a and fig 3b do not match

Response:

Results reported in Figure 3a and Figure 3b do not match since: 1) the gel reported in Figure 3a is a representative zymographic gel referred to the analysis of a single sample per group, while the data of Figure 3b represent the mean+ SD of all the samples analyzed (n=7) for each group of rats. 2) As reported in the sentence of pag. 4, lines 133-135 (paragraph 2.3) the quantitative analysis of MMP-2 and MMP-9 levels, calculated after scanning densitometry and computerized analysis, represents the sum of the pro and activated forms of the two gelatinases.  

Comment: Please fix the figure caption for Figure 2 and Figure 3.

Response:

We have fixed the caption of Figure 2 and Figure 3

Comment: In Figure 3, 4, 5, and 6, the error bars for the control samples should be presented.

Response:

In Figure 3,4, 5 and 6 the error bars for control samples have been added.

Round 2

Reviewer 1 Report

Comments and Suggestions for Authors

The authors carried out all the necessary revision comments. Recommend for publication

Author Response

We thank the reviewer 

Reviewer 2 Report

Comments and Suggestions for Authors

1. Based on the body weight in the Figure 2 and the locmotor status of the Mn and Mn+RSV (Table 2) after 3-, and 4-week treatmnets, the systemic toxicities rather than the inflammations in rats were observed. 

2. Authors should check the protein expressions of Nramps and DMT-1, transferrin for transport proteins of Mn^2+ and Mn^3+.

Author Response

Reviewer #2

Comment: Based on the body weight in the Figure 2 and the locomotor status of the Mn and Mn+RSV (Table 2) after 3-, and 4-week treatmnets, the systemic toxicities rather than the inflammations in rats were observed. 

Response:

We agree with the reviewer that the changes in body weight and locomotor status, observed in rats after 3-, and 4 weeks of treatments with Mn or Mn+RSV, could be due to systemic toxicities rather than to the inflammatory status.

However, we would point out that the results reported in Table 1 and Fig. 2 are referred to the evaluation of the clinical status of rats during the treatment and in this regard, we have never stated that the weight loss and the changes in locomotor status were due to an inflammatory status, but rather to a systemic toxicity. Indeed, in the Discussion Section (Page 7, lines 229-232), to comment on our results, we reported the sentence: "As already suggested by other authors the progressive weight loss in Mn-treated rats could be associated with Mn- induced metabolic and biochemical dysfunctions such as deficiency in energy metabolism and/or alterations in the functionality of hypothalamus, which is responsible for the control of body weight [24,40-33-36]".

Comment: Authors should check the protein expressions of Nramps and DMT-1, transferrin for transport proteins of Mn^2+ and Mn^3+.

Response:

We thank the reviewer for his suggestion, but, unfortunately, we cannot satisfy his request because we do not have sufficient samples to carry out immunohistochemistry or immunoblotting experiments since most of the brain tissue was used for the purification of MMP-2 and MMP-9 by affinity chromatography.

However, numerous studies have shown an increase in DMT-1 levels in different brain areas of rats treated with Mn. To underline these data, in the new version of the manuscript we have added, in the Introduction Section (pag. 1-2, lines 45-48), the following sentence: “Among these importers, DMT-1 is the primary transport system for Mn2+ while TfR-dependent transport is the primary transporter for Mn3+.  Experimental studies have demonstrated the increase of DMT-1 protein levels in different rat brain regions, especially under low Fe conditions, with concurrent increase in Mn levels”, and a new reference (ref 10).

Furthermore, reference 10 of the previous version has been replaced with a new reference (ref 11 in the new version) in which the authors focus their attention on studies related to mechanisms underlying Mn import and export and their function and roles in Mn-induced neurotoxicity.

Reviewer 3 Report

Comments and Suggestions for Authors

This revised manuscript has been improved based on the comments from the reviewers. However, still one of my concern is not cleared.

Since the authors have tested only 7 rats, it is not a good way to present the numbers of rats in %. The number of rats is not enough.

Author Response

Reviewer #3

Comment: Since the authors have tested only 7 rats, it is not a good way to present the numbers of rats in %. The number of rats is enough.

Response:

According to the request of the reviewer we have now delete the percentages in Table 1. Results (pag 3, lines 91-92) and caption of Table 1 have been corrected accordingly.

Round 3

Reviewer 2 Report

Comments and Suggestions for Authors

It is acceptable in the present form.